# The effects of high-heeled shoes on gait parameters in healthy adult women

**Kyoma Tanigawa, Hiroki Shimizu**⬥, **Anuradhi Bandara**⬥, **Misa Toyota, Shota Suzuki, Momoko Nagai-Tanima**⬥, **Tomoki Aoyama**⬥*

Department of Physical Therapy, Human Health Sciences, Graduate School of Medicine, Kyoto University, Kyoto, Japan

* aoyama.tomoki.4e@kyoto-u.ac.jp

## Abstract

### Background

Walking is essential in daily life, and footwear type significantly affects walking patterns. High-heeled shoes increase the risk of knee osteoarthritis and falls in women. Traditional studies often use treadmills or unfamiliar footwear, which may not reflect daily walking. This study investigated the impact of high-heeled shoes on walking parameters in healthy adult women using in-shoe motion sensors.

### Methods

Seventeen healthy adult women without pain during walking participated. They walked for 6 minutes along a 30-meter corridor wearing high-heeled shoes and sneakers. Walking data were recorded using an in-shoe motion sensor system every 2 minutes. The average of three valid consecutive steps was calculated automatically. Statistical analysis compared the mean walking parameters between the high-heel and sneaker groups.

### Results

The high-heel group showed significantly reduced foot clearance, stride length, peak plantar angle in dorsiflexion, and peak plantar angle in plantarflexion, and significantly greater toe-out angle in spatial parameters. Spatiotemporal parameters revealed significantly reduced walking speed and maximum swing phase speed in the high-heel group. No significant differences were observed in temporal parameters between the groups.

### Conclusions

Since this study collected gait data under conditions similar to daily life, it provides data suitable for practical applications and may contribute to future research

**Data availability statement:** "All relevant data are within the paper and its Supporting Information files."

**Funding:** The author(s) received no specific funding for this work.

**Competing interests:** The authors have declared that no competing interests exist.

evaluating everyday gait. Additionally, future studies should include a broader range of participants and incorporate measurement devices capable of capturing hip and knee joint movements, providing a more comprehensive evaluation of the effects of high-heeled shoes on gait in healthy adult women.

## Introduction

Walking is a crucial means of transporting the body safely and efficiently [1]. Additionally, it has beneficial effects on both physical and mental health, enhances human quality of life (QOL), and plays a significant role in the prevention and treatment of various diseases [2]. As such, walking is indispensable in daily life. It is also known that walking is influenced by various external factors [3].

One of these external factors is footwear. Shoes come in various types and have multiple functions, including protecting the feet from external environments, assisting walking functions, and adapting to foot deformities. These functions are related to the occurrence of various conditions such as falls, osteoarthritis, and back pain [4]. Numerous studies have investigated the impact of shoes on walking parameters [5,6]. For example, research comparing walking barefoot and wearing athletic shoes by Shakoor et al. found that wearing athletic shoes decreases step length, increases the number of steps, and reduces joint angles of the lower limbs [5]. Thus, research on how shoes affect walking is of great importance [7].

One type of shoe that warrants attention is high-heeled shoes. High-heeled shoes are characterized by heels higher than the forefoot, narrow toe boxes, and curved soles [8]. High-heeled shoes have been widely worn by women worldwide for centuries, with 37% to 69% of women wearing them regularly and for an average of 1–8 hours per day [8]. Despite their regular use, high-heeled shoes are associated with medical conditions, such as osteoarthritis, and can cause frequent falls [9]. The incidence of such injuries doubled from 7.1% to 14.2% between 2002 and 2012 [10]. Therefore, it is crucial to investigate the impact of high-heeled shoes on the walking parameters of healthy adult women [7]. Several previous studies have examined this effect [9,11–13]. For instance, it has been shown that wearing high-heeled shoes does not change the temporal walking parameters such as stance time and cadence [11]. Additionally, a study by Barkema et al. noted that the higher the heel height, the greater the maximum knee adduction moment, indicating a relationship between heel height and knee osteoarthritis [9].

Research on the effects of high-heeled shoes on gait parameters has significantly progressed. However, many evaluations have been conducted using a treadmill with participants walking at a constant speed [14]. Treadmill walking occurs under flat, steady-speed conditions, which differs from everyday walking environments [15]. For example, because a treadmill maintains a fixed speed, participants cannot walk at their natural pace, potentially suppressing their typical gait patterns [16,17]. Additionally, on a treadmill, individuals often need to adjust their posture and gait mechanics to match the treadmill's motion in order to maintain visual and balance cues, which is known to influence gait biomechanics [15]. In certain studies, participants wore

unfamiliar shoes that may not accurately reflect their usual gait patterns in daily life [11,14]. Shoes that are unfamiliar or uncomfortable for an individual often do not fit well with the person's foot shape or gait style, potentially affecting the gait biomechanics [18]. Specifically, when the shoe's size or shape does not align with the foot, it may cause foot pain and fatigue, while also reducing stability and increasing the risk of falls during walking [19]. Therefore, conditions different from those in daily life may alter an individual's natural gait patterns and may not accurately reflect habitual walking [11].

As a more suitable evaluation method, Inertial Measurement Units (IMUs) can be used. IMUs can estimate three-dimensional movement from angular velocity and acceleration [20,21]. Due to their low cost, small size, and ease of use, they have been widely used to evaluate daily life walking in recent years [22]. By using the In-Shoe Motion Sensor system (IMS), which measures the acceleration and angular velocity of the feet using IMUs and calculates walking parameters, it is possible to obtain walking data comparable to daily life more accurately than conventional studies [23].

Therefore, this study aimed to evaluate the impact of high-heeled shoes on gait parameters in healthy adult women using the IMS system. Unlike previous studies that typically used treadmill walking at a constant speed or required participants to wear unfamiliar shoes, this study allowed participants to walk in their own footwear that fitted their feet and that they were accustomed to, under conditions similar to those of daily life. This approach enabled us to capture a more natural gait pattern, closely resembling everyday walking, and to more accurately reflect the effects of high-heeled shoes on gait parameters.

## Materials and methods

### Study design

This was an intervention study.

### Participants

The participants in this study were healthy adult women enrolled at Kyoto University. The inclusion criteria were individuals aged 18 years and older who could walk for at least 6 minutes without rest. To minimize the influence of unfamiliarity with high-heeled shoes on the results, we selected participants with extensive experience wearing high heels in various settings, such as work, social events, shopping, or formal occasions, based on previous studies [14]. The exclusion criteria were individuals with neurological or orthopedic conditions that could affect walking function. The purpose of the study was thoroughly explained to the participants, and their consent was obtained prior to measurement.

### Ethics statement

This study was approved by the Kyoto University Medical Ethics Committee (Approval number: R3664-3). Prior to study initiation, written informed consent was obtained from all participants in accordance with the ethical standards and guidelines.

### Equipment

The IMS System (A-RROWG, NEC Corporation, Tokyo, Japan) was used for the measurements. The IMS is a device used for gait analysis and consists of key components including an ARM Cortex-M4F MCU (nRF52832 by Nordic Semiconductor). This component features a 64 MHz CPU, 64 KB of RAM, and 512 KB of ROM. Additionally, it includes a Bosch IMU (BMI160) for precise motion tracking, an ABLIC EEPROM (S-24C32C, 32K-bit) for storage, and an Epson real-time clock (RX8130CE). This IMS maintains measurement accuracy by applying real-time correction through two algorithms: the Three-dimensional zero-velocity-update (3D ZUPT) algorithm and the On-line stride segmentation algorithm. The on-line stride segmentation algorithm enables stable foot-flat detection using the plantar angle, ensuring precise stride segmentation. Additionally, the 3D ZUPT algorithm effectively compensates for drift errors, allowing for accurate stride parameter computation within the IMS device [24]. Due to the implementation of these algorithms, no manual calibration

of the IMU was conducted in this study. This device performs gait measurements at 2-minute intervals, capturing gait parameters on both sides. The system detects the gait cycle during flat walking, operates for 1 min, and then enters sleep mode to conserve energy. During this 1-minute period, it detects vibrations and identifies stable walking patterns, and upon detecting the start of a gait cycle, it measures the parameters for three steps within the following 9 s. These steps do not need to be consecutive; if the three steps are successfully measured within a specified time, the average value is recorded. If fewer than three steps were measured, the average value was not recorded. Additionally, this measurement process can be repeated up to three times within a 1-minute window; if all three attempts fail, the measurement for that period is considered unsuccessful, and no data is recorded [24–26].

The following 16 parameters are calculated: stride length, maximum (peak) dorsiflexion angle, maximum (peak) plantar flexion angle, foot clearance, foot height, circumduction, toe-in/out angle, roll angle of heel contact, roll angle of toe-off, walking speed, peak swing angular velocity, maximum speed during swinging phase, stance time, swing time, pushing time, and cadence. Table 1 below, disclosed by NEC Corporation, categorizes the definitions of each parameter into spatial, spatiotemporal, and temporal parameters. These parameters were validated in previous studies for accuracy or were derived from validated parameters [24–26]. High-heeled shoes were defined as "footwear with a heel elevated higher than the forefoot, a narrow toe box, and a curved sole" [11], while sneakers were defined as "light footwear that has a top made of cloth and a bottom made of rubber." Based on previous studies that used shoes with a heel height of 3 cm or more as high-heeled shoes [27,28], the present study defined high-heeled shoes as those with a heel height of 3 cm or greater. Accordingly, the habitual shoes used in this study refer to the footwear that the participants considered to be the ones they wore most frequently during daily activities, such as commuting and shopping.

## Procedure

The measurements were conducted in a 30-meter corridor at the Faculty of Human Health Sciences building, at Kyoto University. First, we verified whether the shoes worn by the participants met the aforementioned criteria. Subsequently, we measured the age (years), height (cm), body mass(kg), size of the shoes(cm), heel height (cm) of the high-heeled shoes, and mass (g) of each shoe. Prior to data collection, participants walked 5 m with insoles equipped with an IMS inserted into

Table 1. Definitions of each walking parameter.

| Classification | Parameters | Definition |
|---|---|---|
| Spatial parameters | Stride length (cm) | Distance from the starting point to end point of the foot trajectory |
| | Maximum (peak) dorsiflexion angle (degree) | Peak foot-sole angle in dorsiflexion |
| | Maximum (peak) plantar flexion angle (degree) | Peak foot-sole angle in plantarflexion |
| | Foot clearance (cm) | Maximum heel height of the foot trajectory |
| | Foot height (cm) | Maximum vertical height of the foot trajectory |
| | Circumduction (cm) | Displacement in the medial–lateral direction during the swing phase |
| | Toe-in/out angle (degree) | Average adduction/abduction angle during the swing phase |
| | Roll angle of heel contact (degree) | Plantar roll angle when the heel touches the ground |
| | Roll angle of toe-off (degree) | Plantar roll angle when the toe leaves the ground |
| Spatiotemporal parameters | Walking speed (km/h) | Stride length divided by stride time |
| | Peak swing angular velocity (degree/s) | Peak angular velocity of the foot during the swing phase |
| | Maximum speed during swinging phase (km/h) | Maximum forward speed of the swinging leg during the swinging phase |
| Temporal parameters | Stance time(s) | Amount of time that the foot is on the ground during the gait cycle |
| | Swing time(s) | Amount of time that the foot is off the ground during the gait cycle |
| | Pushing time (s) | Time between the heel contact and toe leaving the ground. |
| | Cadence (steps/min) | Number of steps per min |

their shoes to ensure that they experienced no discomfort during walking. For data collection, Participants placed a smartphone, used for data recording, in either pocket of their pants, as there were no restrictions on which side to use. They were instructed to begin walking from one end of the corridor, turn around at the opposite end, and continue this back-and-forth walking for 6 min. The measurements were conducted twice: once in sneakers and once in high-heeled shoes, with the order of conditions determined using Excel's random function. Participants were given the option to take a break between trials upon request. The observer stood in the middle of the pathway during the test to ensure that the participants walked normally and without safety issues, providing updates on the remaining time every minute. If no data were collected, the 6-minute walking test was repeated. In this study, real-life conditions refer to natural walking situations that closely resemble daily life, without the use of treadmills or experimental footwear. Natural gait refers to an individual's inherent walking pattern under conditions without external constraints or artificial instructions, and without any intentional modification of their gait.

### Data analysis

Statistical analyses were performed using the JMP PRO16 software (SAS Institute). First, the Shapiro-Wilk test was conducted to determine whether each dataset followed a normal distribution. For data that were normally distributed, a paired t-test was used, because the same participants were measured under different conditions (high-heeled shoes and sneakers). For data that did not follow a normal distribution, the Wilcoxon signed-rank test, suitable for paired datasets, was applied. The group wearing high-heeled shoes was referred to as the "high-heeled group" (HG) and the group wearing sneakers as the "sneaker group" (SG). Data are presented as mean standard deviation ± standard deviation (SD). The effect size was evaluated using Cohen's d to assess the magnitude of the differences between groups, with thresholds defined as Small (0.2–0.49), Medium (0.5–0.79), and Large (≥ 0.8) [29]. The statistical significance level for this study was set at less than 5%.

## Results

### A. Physical characteristics (Table 2)

There were 17 participants in the study, with no dropouts or adverse events reported. The mean values were as follows: age, 23.4 years; height, 161.78 cm; body mass, 53.46 kg; BMI, 20.38; size of shoes, 24.15 cm; mass of high-heeled shoes, 201.76 g; mass of sneakers, 265.00 g; heel height of high-heeled shoes, 4.93 cm (Table1).

### B. Comparison of walking parameters between high-heeled and sneaker shoes (Table 3)

From the Table 3, in the comparison between the high-heeled group and the sneaker group, spatial parameters showed that the high-heeled group had a significantly shorter stride length, reduced peak dorsiflexion angle of the foot, reduced

Table 2. Basic information of participants.

| Characteristics | Mean±SD(N=17) |
| --- | --- |
| Age(years) | 23.40±1.23 |
| Height(cm) | 161.78±6.14 |
| Body Mass(kg) | 53.46±5.61 |
| BMI(kg/m²) | 20.38±1.09 |
| Size of shoes(cm) | 24.15±1.01 |
| Mass of heel shoes(g) | 201.76±37.12 |
| Mass of sneakers(g) | 265.00±59.48 |
| Heel height of heel shoes(cm) | 4.93±1.17 |

SD, standard deviation; BMI, body mass index.

**Table 3. Comparisons of the mean values of each parameter in each category.**

| Classification | Parameters | HG | SG | P-value | Effect Size (Cohen's d) |
|---|---|---|---|---|---|
| **Spatial parameters** | Stride length (cm) | 130.42 ± 18.14 | 143.88 ± 16.09 | 0.0006* | −1.09 |
| | Maximum (peak) dorsiflexion angle (degree) | 24.21 ± 5.25 | 29.88 ± 5.13 | 0.0017* | −0.94 |
| | Maximum (peak) plantar flexion angle (degree) | 62.33 ± 11.15 | 78.31 ± 6.49 | <0.0001* | −1.47 |
| | Foot clearance (cm) | 21.17 ± 3.31 | 24.05 ± 2.15 | 0.0005* | −1.09 |
| | Foot height (cm) | 13.33 ± 2.40 | 15.70 ± 1.76 | 0.0023* | −0.90 |
| | Circumduction (cm) | 4.10 ± 1.58 | 3.72 ± 2.01 | 0.5397 | 0.15 |
| | Toe-in/out angle (degree) | 12.05 ± 8.82 | 10.14 ± 7.97 | 0.0476* | 0.54 |
| | Roll angle of heel contact (degree) | 5.21 ± 4.86 | 3.71 ± 5.70 | 0.2343 | 0.30 |
| | Roll angle of toe-off (degree) | −2.05 ± 6.17 | −1.36 ± 4.52 | 0.5178 | −0.17 |
| **Spatiotemporal parameters** | Walking speed (km/h) | 4.61 ± 0.68 | 5.08 ± 0.66 | 0.0005* | −1.09 |
| | Peak swing angular velocity (degree/s) | 531.64 ± 95.96 | 565.64 ± 75.54 | 0.0932 | −0.45 |
| | Maximum speed during swinging phase (km/h) | 15.15 ± 1.65 | 15.97 ± 1.59 | 0.0085* | −0.75 |
| **Temporal parameters** | Stance time(s) | 0.62 ± 0.05 | 0.62 ± 0.04 | 0.8271 | −0.06 |
| | Swing time(s) | 0.41 ± 0.04 | 0.40 ± 0.01 | 0.3225 | 0.03 |
| | Pushing time (s) | 0.19 ± 0.03 | 0.20 ± 0.02 | 0.6805 | −0.13 |
| | Cadence (steps/min) | 117.22 ± 6.41 | 116.92 ± 5.69 | 0.8030 | 0.06 |

SD, standard deviation. *p < 0.05.

Effect size (Cohen's d): Small = 0.2–0.49, Medium = 0.5–0.79, Large ≥ 0.8.

peak plantar flexion angle of the foot, and significantly greater toe angle compared to the sneaker group. Regarding spatiotemporal parameters, the high-heeled group exhibited significantly reduced walking speed and maximum speed during swinging phase. There were no significant differences observed between the groups in temporal parameters.

## Discussion

This study aimed to evaluate the effect of high-heeled shoes on gait parameters in healthy adult women using an IMS system. In previous studies, participants often wore unfamiliar shoes and measurements were commonly conducted on a treadmill at a constant speed, which may not accurately reflect natural gait patterns in real-life settings. In contrast, this study utilized a compact and user-friendly IMS to collect gait data while participants wore their own familiar shoes. This approach allowed for accurate collection in a manner closer to daily life conditions, providing foundational data for assessing the effects of footwear under more realistic circumstances.

As a result, in terms of spatial parameters, the high-heeled group showed significantly shorter stride length, reduced maximum dorsiflexion angle, reduced maximum plantar flexion angle, and significantly greater toe-out angle compared to the sneaker group, with effect sizes ranging from moderate to high. Additionally, in spatiotemporal parameters, the high-heeled group exhibited significantly reduced walking speed and maximum speed during the swinging phase, with large effect sizes. However, no significant differences were observed between the groups in temporal parameters.

Regarding spatial parameters such as walking speed, stride length, and swing speed, all were significantly reduced in the high-heeled group, with large to moderate effect sizes: Cohen's d = −1.09 for stride length and walking speed, and Cohen's d = −0.75 for swing speed. These results strongly suggest that wearing high-heeled shoes has a substantial impact on propulsion and walking efficiency. Generally, wearing high heels narrows the base of support during walking, prompting individuals to adopt a safer and more stable gait pattern [30,31]. In a previous study [24], the RMSE for stride length and walking speed based on measurements taken over three steps was reported as 0.042 and 0.071, respectively. In contrast, the differences between the HG and SG in the present study were 0.1346 for stride length and 0.47

for walking speed, both of which substantially exceeded the reported RMSE values. Therefore, these differences are considered to exceed the range of measurement error and can be interpreted as practically meaningful and statistically significant group differences. Additionally, when wearing high heels, the reduced base of support during walking leads to increased muscle activity to maintain balance, which in turn tends to increase energy expenditure. To minimize this increase in energy expenditure, it has been shown that walking speed and stride length are reduced as a strategy [32,33]. In addition to these stable and efficient walking strategies, previous studies have shown that wearing high heels brings the ground reaction force closer to the ankle joint, and the curved shape of the shoe sole causes the ankle to remain in a plantarflexed position [11]. These factors can shorten the length of the triceps surae muscle and Achilles tendons. Consequently, compared to sneakers or barefoot walking, ankle plantarflexion moments decrease, and the effect of the stretch-shortening cycle on the muscle-tendon complex is reduced [34,35]. The stretch-shortening cycle is a process in which muscles and tendons quickly stretch and then shorten, store, and release elastic energy, and plays a crucial role in efficient force production [35]. However, the shortened state of the muscle-tendon complex induced by wearing high heels is thought to impede this function, leading to decreased energy efficiency and a weaker push-off force. As a result, the calculated parameters were significantly different between the two groups.

Similarly, foot clearance and foot height were significantly reduced in the high-heeled group. A very large effect size for foot clearance (Cohen's d = −1.09) and a large effect size for foot height (Cohen's d = −0.90) strongly suggest that foot clearance was more reduced in the high-heeled group. In a previous study [24], the RMSE for foot height was reported to be 0.017. In contrast, the difference between the HG and SG observed in this study was 0.0464, which substantially exceeds the RMSE. Therefore, the difference in foot clearance can be considered to exceed the margin of measurement error and represents a practically and statistically significant intergroup difference. As noted above, wearing high-heeled shoes is suggested to reduce the push-off force of the ankle. Additionally, previous studies investigating the role of the ankle in walking have shown that ankle push-off contributes to up to 50% of the power needed for leg swinging during the terminal stance and early swing phases [36–38]. This reduction in ankle push-off may decrease the leg swing speed, which is essential for the double-pendulum effect during walking [39]. In the double-pendulum effect, an increased hip swing speed results in the relative lifting of the lower leg, consequently increasing foot clearance. This occurs as the hip and knee joints work in coordination, with the hip's movement influencing the lower leg's motion. Specifically, a rapid change in the hip angle causes the lower leg to rise, allowing the foot to clear obstacles before making contact with the ground. In the heel group, however, a reduction in leg swing speed led ~~leads~~ to insufficient knee flexion in the double-pendulum mechanism, causing the lower leg to remain relatively low and potentially resulting in reduced foot clearance.

In fact, prior research comparing joint angles in the sagittal plane under different heel heights at a constant walking speed found that while hip flexion during the swing phase remained unaffected by heel height, the knee flexion angle significantly decreased as heel height increased [40–42]. Moreover, a decrease in the foot clearance height can increase the risk of tripping over small obstacles or uneven surfaces while walking, potentially leading to falls [43]. High-heeled shoes also tilt the toes downward and elevate the heel, reducing the contact area of the foot with the ground, which may further destabilize balance. This combination of reduced foot clearance and a limited base of support likely exacerbates the risk of falls [14].

Thus, the high-heeled group showed significantly reduced stride length, reduced peak dorsiflexion angle of the foot, reduced peak plantar flexion angle of the foot, reduced walking speed, and swing phase maximum velocity, while the toe angle during the swing phase was significantly greater. This change demonstrated a moderate effect size (Cohen's d = 0.54), suggesting that individuals wearing high heels may adjust foot positioning to enhance stability. A greater toe angle indicates a larger ankle abduction angle. Studies investigating the influence of heel height on the anterior-posterior joint moments of the reduced limb have shown that as heel height increases, knee internal rotation moments increase, potentially increasing the load on the medial compartment of the knee [9,44]. Conversely, walking with toes pointing outward reduces knee internal rotation moments and decreases the load on the medial compartment [45]. In this study, our

data suggest that the high-heeled group adopted a walking strategy with toes pointing outward to reduce the load on the medial compartment of the knee. Additionally, since knee joint movements during stance and toe direction were not measured in this study, further experiments are necessary. In a previous study [24], the RMSE for toe angle during the swing phase was reported to be 2.071. In contrast, the difference between the HG and SG observed in this study was 1.91, which is approximately the same as the RMSE. Therefore, it is possible that the observed difference in toe angle may not be entirely free from the influence of measurement error.

There are limitations in this study; the participants were limited to healthy adult women from the same university, which restricts the generalizability of our findings. To apply these results to other age groups, sexes, or individuals with specific health conditions, further studies with more diverse participants are necessary. Second, the sample size in this study was limited to 17 participants, which poses constraints on statistical robustness and the power to detect effects. In particular, to adequately assess the impact of individual differences, it is essential to include a larger number of participants. Future studies should increase the sample size to enhance statistical robustness and improve the power to detect effects. Third, we were unable to capture the movements of the hip and knee joints, making it challenging to fully assess how these joints affect gait mechanics, particularly balance and posture control. Specifically, it remains unclear to what extent high-heeled shoes stress on these joints under near-daily conditions and how this stress might increase the risk of falls or lead to long-term joint disorders. Future research should incorporate devices that measure the movements of these joints to provide a more comprehensive evaluation of the impact of different footwear types on gait and balance. Finally, for the toe angle, the difference between groups was comparable to the RMSE, making it difficult to conclusively determine that the observed difference was statistically meaningful. Therefore, future studies should adopt more precise measurement techniques to verify the reliability of this difference.

## Conclusions

This study investigated the effects of high-heeled shoes on walking parameters in healthy adult women using the IMS. As a result, in terms of spatial parameters, the high-heeled group showed significantly shorter stride length, reduced maximum dorsiflexion angle, reduced maximum plantar flexion angle, and significantly greater toe-out angle compared to the sneaker group, with effect sizes ranging from moderate to high. Additionally, in spatiotemporal parameters, the high-heeled group exhibited significantly reduced walking speed and maximum speed during the swinging phase, with large effect sizes. However, no significant differences were observed between the groups in temporal parameters. These findings suggest that wearing high-heeled shoes may have an impact on gait instability and efficiency, potentially increasing the risk of falls and musculoskeletal strain. Future studies should include a more diverse population, such as older adults or individuals with joint disorders, and incorporate motion capture systems or wearable sensors that can measure reduced limb joint dynamics. These additional studies would allow for a comprehensive understanding of how footwear affects not only the ankle but also hip and knee joint movements across a wider population. Because this study collected gait data under conditions similar to those of daily life, it provides data suitable for practical applications and may contribute to future research focused on evaluating gait in everyday scenarios.

## Acknowledgments

We express our gratitude to all participants, instructors, and laboratory members for their cooperation and valuable input throughout this study. We also thank Editage (www.editage.com) for English editing.

## Author contributions

**Conceptualization:** Kyoma Tanigawa, Hiroki Shimizu, Anuradhi Bandara, Misa Toyota, Shota Suzuki, Momoko Nagai-Tanima.

**Data curation:** Kyoma Tanigawa, Hiroki Shimizu, Anuradhi Bandara, Misa Toyota, Shota Suzuki, Momoko Nagai-Tanima.

 

**Formal analysis:** Kyoma Tanigawa, Hiroki Shimizu, Anuradhi Bandara, Misa Toyota, Shota Suzuki, Momoko Nagai-Tanima.

**Investigation:** Kyoma Tanigawa.

**Supervision:** Tomoki Aoyama.

**Validation:** Hiroki Shimizu.

**Writing – original draft:** Kyoma Tanigawa.

**Writing – review & editing:** Kyoma Tanigawa, Hiroki Shimizu, Anuradhi Bandara, Misa Toyota, Shota Suzuki, Momoko Nagai-Tanima, Tomoki Aoyama.

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
