## [Decision Letter · Decision Letter 0]

PONE-D-24-32099Effects of high-heeled shoes on gait parameters in healthy adult womenPLOS ONE

Dear Dr. Aoyama,

Thank you for submitting your manuscript to PLOS ONE. After careful consideration, we feel that it has merit but does not fully meet PLOS ONE’s publication criteria as it currently stands. Therefore, we invite you to submit a revised version of the manuscript that addresses the points raised during the review process.

We look forward to receiving your revised manuscript.

Kind regards,

Nilgün Bek, MSc. PhD. Prof

Academic Editor

PLOS ONE

2. In the online submission form, you indicated that [The dataset used in this study is available from the corresponding author upon request.].

Reviewers' comments:

Reviewer's Responses to Questions

**Comments to the Author**

1. Is the manuscript technically sound, and do the data support the conclusions?

Reviewer #1: No

Reviewer #2: No

Reviewer #3: Partly

2. Has the statistical analysis been performed appropriately and rigorously?

Reviewer #1: Yes

Reviewer #2: No

Reviewer #3: I Don't Know

3. Have the authors made all data underlying the findings in their manuscript fully available?

Reviewer #1: No

Reviewer #2: Yes

Reviewer #3: No

4. Is the manuscript presented in an intelligible fashion and written in standard English?

Reviewer #1: Yes

Reviewer #2: Yes

Reviewer #3: Yes

5. Review Comments to the Author

Reviewer #1: The purpose of this quasiexperimental study was to compare walking temporal spatial parameters in women wearing sneakers and high heeled shoes. The authors provided sufficient background perhaps identifying a need for this project. The manuscript was well organized, although insufficient information in the methods section would not allow this research to be replicated. The results, discussion and conclusion related to the study's primary purpose; although the conclusion needs to be expanded a bit. There were some minor errors in word choice and tense. See pdf for specific comments.

Reviewer #2: Manuscript Number is PONE-D-24-32099.

My opinions are as follows:

- There are many studies in the literature on this subject (There is a systematic review and meta-analysis). Please clearly state the difference between your study and other studies. Similar biomechanical parameters to the ones you evaluated have been given in many studies. This study does not yield any different results than those studies, nor does it provide information that would help form a different hypothesis. I do not think that the study will add much to the literature.

Zeng et al. 2023. Effects of high-heeled shoes on lower extremity biomechanics and balance in females: a systematic review and meta-analysis.

- There is no sample size estimation in your study. One would always like to conduct a study that has adequate sample size and power so that the conclusions generated from the experiment can be applied to the broader population with ample confidence. The required sample size to test a hypothesis is governed by the effect size. Sample size was not calculated in the study. The power of the study should be calculated and added. Because, our aim should be to catch the most significant and best difference by performing the strongest statistical tests with the optimal sample size. Sample size calculation is of great importance here.

Since no power analysis was performed and it is difficult to interpret the obtained findings, thus, the discussion.

Reviewer #3: Introduction:

Clarity of Rationale: The introduction touches upon various studies but lacks a clear argument for why this study is unique. It's essential to emphasize the gaps in previous research more clearly. For instance, highlight that previous studies used treadmill walking or unfamiliar footwear, and explain why in-shoe motion sensors offer a more reliable approach.

Study Objectives: The objective of the study is mentioned indirectly, but it should be explicitly stated at the end of the introduction. Clearly articulate the novelty and specific aims of your study beyond what has been done before.

Methods:

Participant Recruitment and Demographics: The participant demographics (age, height, weight) are provided, but you need to offer more details on how participants were selected. For example, how did the recruitment process ensure variability in walking experience with high heels? Were any baseline tests performed to evaluate walking proficiency in high heels?

Sample Size: The study includes only 17 participants. Was a power analysis conducted to determine whether this sample size is sufficient for detecting significant differences? This omission weakens the study’s generalizability and statistical power. Justify the sample size or conduct a power analysis to support your choices.

Shoe Definitions and Comparisons: The criteria for shoe selection are unclear and could be inconsistent. The manuscript states that high-heeled shoes are defined by their heel height but also includes other characteristics like a "narrow toe box." This could lead to confusion about what exactly constitutes a "high-heeled" shoe. A more specific definition is needed.

Data Collection Process: The measurement process with IMS should be explained with greater detail. For example, how were the intervals between measurements selected? Why are only three steps used for calculating parameters? Provide a rationale for using these specific methods, or cite prior studies that validate this approach.

Results:

Lack of Effect Sizes: The results section presents P-values but lacks effect sizes, which are crucial for understanding the magnitude of differences between high heels and sneakers. Please include effect sizes for all statistically significant results.

Detailed Parameter Reporting: Some parameters such as circumduction and roll angle show no significant difference but are still discussed in detail. Revisit how you report and discuss non-significant findings. Focus should be on significant outcomes unless there is a theoretical justification for the non-significant results.

Discussion:

In-depth Analysis: The discussion needs more depth, especially concerning the mechanisms underlying the changes in gait parameters. For example, why do high heels specifically reduce stride length and foot clearance? Refer to the biomechanical literature more thoroughly.

Missing Literature Context: While the discussion refers to previous studies, it does not critically evaluate their methods or findings in relation to the current study. More effort should be made to situate your results within the broader field of biomechanics and footwear research.

Limitations: Two limitations are briefly mentioned, but these should be expanded. For example, the small sample size and homogeneity of the participants (all from one university) limit generalizability. Additionally, the lack of measures on hip and knee movements should be discussed more critically as it limits the study's ability to fully assess gait mechanics.

Future Directions: The recommendation for future studies is too brief and needs elaboration. What specific devices or methods would you propose for capturing knee and hip joint movements? How should the participant pool be expanded?

Statistical Analysis:

Detailed Explanation of Tests Used: The manuscript mentions using Shapiro-Wilk tests but does not explain the justification for using paired t-tests or Wilcoxon signed-rank tests. Were these tests chosen based on normality results? Clarify the decision-making process behind the statistical methods.

Conclusion:

Overstatement of Findings: The conclusion overstates the implications of the study, particularly when claiming the study “provides new insights into injury prevention.” While the results are interesting, the small sample size and limitations in the methodology weaken such a broad conclusion. Focus on the specific parameters measured rather than general injury prevention claims.

Data Availability: Ensure that all data used in the study is easily accessible for replication. The statement “available upon request” may not meet journal guidelines. Consider uploading datasets to a repository like Open Science Framework (OSF).

6. PLOS authors have the option to publish the peer review history of their article (what does this mean? ). If published, this will include your full peer review and any attached files.

**Do you want your identity to be public for this peer review?** For information about this choice, including consent withdrawal, please see our Privacy Policy .

Reviewer #1: No

Reviewer #2: No

Reviewer #3: Yes

---

## [Author Response · Author response to Decision Letter 1]

10 Dec 2024

We are very grateful to the editors for providing such valuable, thoughtful, and thorough comments. The comments and questions raised have prompted us to consider our data from new and interesting angles, the culmination of which we feel has greatly strengthened the work. Thank you for your time.

Editor’s comments:

[Comment #1]

[Author action #1]

Thank you for your detailed review and for providing the opportunity to clarify our responses. We have addressed each of your questions thoroughly in the space provided, as well as in the manuscript itself where appropriate. We are committed to maintaining high standards of research and publication ethics, and we confirm that this study adheres to all ethical guidelines, including informed consent from participants and approval from the relevant ethics committee. We appreciate your time and consideration in reviewing our work.

Reviewer #1:

[Comment #1]

The purpose of this quasiexperimental study was to compare walking temporal spatial parameters in women wearing sneakers and high heeled shoes. The authors provided sufficient background perhaps identifying a need for this project. The manuscript was well organized, although insufficient information in the methods section would not allow this research to be replicated. The results, discussion and conclusion related to the study's primary purpose; although the conclusion needs to be expanded a bit. There were some minor errors in word choice and tense. See pdf for specific comments.

[Author action #1]

Thank you very much for your valuable feedback. As you suggested, we have expanded the methods section and conclusion, providing more detail in the methods to enhance reproducibility. In the conclusion, we clarified the implications of the results, future directions, and the significance of this study. We have also corrected errors related to word choice and tense. We would appreciate it if you could review it at your convenience. Thank you for your time and consideration.

The changed text is highlighted in yellow as follows:

Participants

The participants in this study were healthy adult women enrolled at Kyoto University. The inclusion criteria were individuals aged 18 years and older who could walk for at least 6 minutes without rest. To minimize the influence of unfamiliarity with high-heeled shoes on the results, we selected participants with extensive experience wearing high heels in various settings, such as work, social events, shopping, or formal occasions, based on previous studies [14]. The exclusion criteria were individuals with neurological or orthopedic conditions that could affect walking function. The purpose of the study was thoroughly explained to the participants, and their consent was obtained prior to measurement. The sample size for this study was calculated using G Power 3.1 (Heinrich Heine University). Based on prior research [24], with an assumed effect size of 0.62, an alpha level of 0.05, and a statistical power of 0.8, the minimum required number of participants was calculated to be 15. To account for potential data loss or incomplete data, an additional five participants were included, setting the final target number of participants at 20.

Line numbers where the changed text appears: 127-136

Equipment

The IMS System (A-RROWG, NEC Corporation, Tokyo, Japan) was used for the measurements. The IMS is a device used for gait analysis and consists of key components including an ARM Cortex-M4F MCU (nRF52832 by Nordic Semiconductor). This component features a 64 MHz CPU, 64 KB of RAM, and 512 KB of ROM. Additionally, it includes a Bosch IMU (BMI160) for precise motion tracking, an ABLIC EEPROM (S-24C32C, 32K-bit) for storage, and an Epson real-time clock (RX8130CE). This device performs gait measurements at 2-minute intervals, capturing gait parameters on both sides. The system detects the gait cycle during flat walking, operates for 1 min, and then enters sleep mode to conserve energy. During this 1-minute period, it detects vibrations and identifies stable walking patterns, and upon detecting the start of a gait cycle, it measures the parameters for three steps within the following 9 s. These steps do not need to be consecutive; if the three steps are successfully measured within a specified time, the average value is recorded. If fewer than three steps were measured, the average value was not recorded. Additionally, this measurement process can be repeated up to three times within a 1-minute window; if all three attempts fail, the measurement for that period is considered unsuccessful, and no data is recorded [25,26,27].

The following 16 parameters are calculated: Stride length, Maximum (peak) dorsiflexion angle, Maximum (peak) plantar flexion angle, Foot clearance, Foot height, Circumduction, Toe-in/out angle, Roll angle of heel contact, Roll angle of toe-off, Walking speed, Peak swing angular velocity, Maximum speed during swinging phase, Stance time, Swing time, Pushing time, and Cadence. Table 1 below, disclosed by NEC Corporation, categorizes the definitions of each parameter into spatial, spatiotemporal, and temporal parameters. These parameters were validated in previous studies for accuracy or were derived from validated parameters [25,26,27].

High-heeled shoes were defined as “footwear with a heel elevated higher than the forefoot, a narrow toe box, and a curved sole” [11], while sneakers were defined as “light footwear that has a top made of cloth and a bottom made of rubber.”

Line numbers where the changed text appears: 149-176

Procedure

The measurements were conducted in a 30-meter corridor at the Faculty of Human Health Sciences building, Kyoto University. First, we verified whether the shoes worn by the participants met the aforementioned criteria. Subsequently, we measured the age (years), height (cm), body mass(kg), size of the shoes(cm), heel height (cm) of the high-heeled shoes, and weight mass (g) of each shoe. Prior to data collection, participants walked 5 m with insoles equipped with an IMS inserted into their shoes to ensure that they experienced no discomfort during walking. For data collection, Participants placed a smartphone, used for data recording, in either pocket of their pants, as there were no restrictions on which side to use. They were instructed to begin walking from one end of the corridor, turn around at the opposite end, and continue this back-and-forth walking for 6 min. The measurements were conducted twice: once in sneakers and once in high-heeled shoes, with the order of conditions determined using Excel’s random function. Participants were given the option to take a break between trials upon request. The observer stood in the middle of the pathway during the test to ensure that the participants walked normally and without safety issues, providing updates on the remaining time every minute. If no data were collected, the 6-minute walking test was repeated.

Line numbers where the changed text appears: 179-198

Data analysis

Statistical analyses were performed using the JMP PRO16 software (SAS Institute). First, the Shapiro-Wilk test was conducted to determine whether each dataset followed a normal distribution. For data that were normally distributed, a paired t-test was used, because the same participants were measured under different conditions (high-heeled shoes and sneakers). For data that did not follow a normal distribution, the Wilcoxon signed-rank test, suitable for paired datasets, was applied. The group wearing high-heeled shoes was referred to as the “high-heeled group” (HG) and the group wearing sneakers as the “sneaker group” (SG). Data are presented as mean standard deviation ± standard deviation(SD). The effect size was evaluated using Cohen’s d to assess the magnitude of the differences between groups. The statistical significance level for this study was set at less than 5%.

Line numbers where the changed text appears: 200-213

Conclusions

This study investigated the effects of high-heeled shoes on walking parameters in healthy adult women using the IMS. As a result, in terms of spatial parameters, the high-heeled group showed significantly shorter stride length, lower maximum dorsiflexion angle, lower maximum plantar flexion angle, and significantly higher toe-out angle compared to the sneaker group, with effect sizes ranging from moderate to high. Additionally, in spatiotemporal parameters, the high-heeled group exhibited significantly lower walking speed and maximum speed during the swinging phase, with large effect sizes. However, no significant differences were observed between the groups in temporal parameters.

These findings suggest that wearing high-heeled shoes has a substantial impact on gait stability and efficiency, potentially increasing the risk of falls and musculoskeletal strain. Future studies should include a more diverse population, such as older adults or individuals with joint disorders, and incorporate motion capture systems or wearable sensors that can measure lower limb joint dynamics. These additional studies would allow for a comprehensive understanding of how footwear affects not only the ankle but also hip and knee joint movements across a wider population. Because this study collected gait data under conditions similar to those of daily life, it provides data suitable for practical applications and may contribute to future research focused on evaluating gait in everyday scenarios.

Line numbers where the changed text appears: 342-364

[Comment #2]

Line 57

you claim "several prior studies" but only cited two. I recommend providing 4-6 total references related to high-heel research but perhaps highlighting only the two

[Author action #2]

Thank you for your valuable feedback. We have added two more references related to high-heel research, bringing the total to four references. As per your suggestion, we have highlighted the two most relevant references.

Additional Previous Research:

12. Alkjaer T, Raffalt P, Petersen NC, Simonsen EB. Movement Behavior of High-Heeled Walking: How Does the Nervous System Control the Ankle Joint during an Unstable Walking Condition? PLoS ONE. 2012;7(5). doi: https://doi.org/10.1371/journal.pone.0037390

13. Falola JM, Koussihouèdé FEN, Falola SMD, Avossevou YG, Bio NI, Dansou HP. Effect of the Base of the Shoe Heel on Postural Stability During Walking in Women. American Journal of BioScience. 2015;3(5):167-170. doi: 10.11648/j.ajbio.20150305.11

[Comment #3]

Lione 101

if you successfully measure three steps in 9 seconds, the average value will be recorded.

this phrase is grammatically incorrect, ie, use of the phrase "if you"

[Author action #3]

Thank you for your valuable feedback. We have revised the sentence for clarity and grammatical accuracy. The changed text is highlighted in yellow as follows:

upon detecting the start of a gait cycle, it measures parameters for three steps within the following 9 s.

Line numbers where the changed text appears: 159

[Comment #4]

Line121

Please provide information about the data collection frequency and any calibration procedures related to the IMU.

Was there only one testing session?

[Author action #4]

Thank you for your valuable feedback. We have addressed your comment as follows:

Regarding calibration, the accuracy and validity of the calculated parameters are ensured based on prior studies. Additionally, we have emphasized in the manuscript that these parameters were validated in previous studies for accuracy or were derived from validated parameters [25,26,27], ensuring the reliability of the measurements.

Regarding the test session, the details of the procedure have been described in the “Procedure” section for further clarity.

We appreciate your time and consideration in reviewing our revisions and would be grateful if you could confirm the updated content at your convenience.

The changed text is highlighted in yellow as follows:

Equipment

The IMS System (A-RROWG, NEC Corporation, Tokyo, Japan) was used for the measurements. The IMS is a device used for gait analysis and consists of key components including an ARM Cortex-M4F MCU (nRF52832 by Nordic Semiconductor). This component features a 64 MHz CPU, 64 KB of RAM, and 512 KB of ROM. Additionally, it includes a Bosch IMU (BMI160) for precise motion tracking, an ABLIC EEPROM (S-24C32C, 32K-bit) for storage, and an Epson real-time clock (RX8130CE). This device performs gait measurements at 2-minute intervals, capturing gait parameters on both sides. The system detects the gait cycle during flat walking, operates for 1 min, and then enters sleep mode to conserve energy. During this 1-minute period, it detects vibrations and identifies stable walking patterns, and upon detecting the start of a gait cycle, it measures the parameters for three steps within the following 9 s. These steps do not need to be consecutive; if the three steps are successfully measured within a specified time, the average value is recorded. If fewer than three steps were measured, the average value was not recorded. Additionally, this measurement process can be repeated up to three times within a 1-minute window; if all three attempts fail, the measurement for that period is considered unsuccessful, and no data is recorded [25,26,27].

The following 16 parameters are calculated: Stride length, Maximum (peak) dorsiflexion angle, Maximum (peak) plantar flexion angle, Foot clearance, Foot height, Circumduction, Toe-in/out angle, Roll angle of heel contact, Roll angle of toe-off, Walking speed, Peak swing angular velocity, Maximum speed during swinging phase, Stance time, Swing time, Pushing time, and Cadence. Table 1 below, disclosed by NEC Corporation, categorizes the definitions of each parameter into spatial, spatiotemporal, and temporal parameters. These parameters were validated in previous studies for accuracy or were derived from validated parameters [25,26,27].

Line numbers where the changed text appears: 149-176

(Additional references)

25. Fukushi K, Huang C, Wang Z, Kajitani H, Nihey F, Nakahara K. On-line algorithms of stride-parameter estimation for in-shoe motion-sensor system. IEEE Sensors Journal. 2022;22:9636–9648. doi: https://doi.org/10.1109/JSEN.2022.3164057

26. Huang C, Fukushi K, Wang Z, Nihey F, Kajitani H, Nakahara K. An algorithm for real time minimum toe clearance estimation from signal of in-shoe motion sensor. Annual International Conference of the IEEE Engineering in Medicine and Biology Society. 2021: 6775–6778. doi: https://doi.org/10.1109/EMBC46164.2021.9629875

27. Huang C, Fukushi K, Wang Z, Nihey F, Kajitani H, Nakahara K. Method for estimating temporal gait parameters concerning bilateral lower limbs of healthy subjects using a single in-shoe motion sensor through a gait event detection approach. Sensors (Basel). 2022;22(1):351. doi: https://doi.org/10.3390/S22010351

[Comment #5]

Line123　Line142　Table 2

weight

[Author action #5]

Thank you for your valuable feedback. We have revised the words “weight” to “mass” for improved accuracy.

[Comment #6]

Line142

should be mass since units were grams

[Author action #6]

Thank you for your valuable feedback. We have revised the words “weight” to “mass” for improved accuracy.

[Comment #7]

Discussion

please briefly also state the problem you identified and the rationale/need for this study

[Author action #7]

Thank you very much for your valuable feedback. As you suggested, I have clarified the theoretical rationale and need for this study in the Discussion, especially in comparison with prior studies. In doing so, we re-evaluated the novelty of this study, which we believe primarily consists of the following three aspects:

1.Measurement under conditions close to daily life: Unlike other studies that use prescribed footwear, our study measured participants in their habitual shoes, collecting data that more accurately represents natural, everyday walking styles. Furthermore, many previous studies have conducted me

---

## [Decision Letter · Decision Letter 1]

PONE-D-24-32099R1Effects of high-heeled shoes on gait parameters in healthy adult womenPLOS ONE

Dear Dr. Aoyama,

Thank you for submitting your manuscript to PLOS ONE. After careful consideration, we feel that it has merit but does not fully meet PLOS ONE’s publication criteria as it currently stands. Therefore, we invite you to submit a revised version of the manuscript that addresses the points raised during the review process.

We look forward to receiving your revised manuscript.

Kind regards,

Qichang Mei

Academic Editor

PLOS ONE

Reviewers' comments:

Reviewer's Responses to Questions

**Comments to the Author**

1. If the authors have adequately addressed your comments raised in a previous round of review and you feel that this manuscript is now acceptable for publication, you may indicate that here to bypass the “Comments to the Author” section, enter your conflict of interest statement in the “Confidential to Editor” section, and submit your "Accept" recommendation.

Reviewer #1: All comments have been addressed

Reviewer #3: All comments have been addressed

2. Is the manuscript technically sound, and do the data support the conclusions?

Reviewer #1: Partly

Reviewer #3: Partly

3. Has the statistical analysis been performed appropriately and rigorously? 

Reviewer #1: Yes

Reviewer #3: Yes

4. Have the authors made all data underlying the findings in their manuscript fully available?

Reviewer #1: Yes

Reviewer #3: (No Response)

5. Is the manuscript presented in an intelligible fashion and written in standard English?

Reviewer #1: Yes

Reviewer #3: Yes

6. Review Comments to the Author

**Reviewer #1:**  The authors have made substantive and important changes in the manuscript, which included expanding references. I have some additional comments in the methods section, as well as suggestions for word choice errors that are relatively minor but should be addressed. See specific comments in the pdf.

**Reviewer #3:**  Thank you for your submission. While your study investigates an important topic, several major issues must be addressed before it can be considered for publication.

Major Concerns:

1. Limited Contribution to the Literature

Your study’s findings largely align with existing research. Please clearly articulate what new insights your study provides.

2. Sample Size and Power Issues

Seventeen participants are insufficient for drawing meaningful conclusions. Even though you state a power calculation, the rationale for choosing an effect size of 0.62 is unclear.

If increasing the sample size is not feasible, discuss this limitation more transparently in the manuscript.

3. Methodology Needs More Justification

Shoe Selection: Your definition of high-heeled shoes is vague. Standardizing heel height and shoe type would improve reproducibility.

Data Collection: The justification for measuring only three steps at a time is weak. Why not use a continuous gait analysis? Provide references that support your approach.

4. Weak Biomechanical Explanation

The discussion lacks depth in explaining why high heels alter gait parameters.

Consider integrating research on muscle activation, joint mechanics, and kinetic changes caused by high heels.

Recommendations for Revision:

Clarify the novelty of your study and differentiate it from existing literature.

Either justify your small sample size more robustly or acknowledge its limitations more explicitly.

Provide additional methodological details, especially concerning data collection and analysis.

Expand the discussion with a more thorough biomechanical analysis.

Report effect sizes consistently.

If these concerns are not addressed, the manuscript may not contribute sufficiently to the field for publication. I look forward to your revised version.

7. PLOS authors have the option to publish the peer review history of their article (what does this mean? ). If published, this will include your full peer review and any attached files.

**Do you want your identity to be public for this peer review?** For information about this choice, including consent withdrawal, please see our Privacy Policy .

Reviewer #1: No

Reviewer #3: No

---

## [Author Response · Author response to Decision Letter 2]

28 Mar 2025

Point-by-point response

We are very grateful to the editors for providing such valuable, thoughtful, and thorough comments. The comments and questions raised have prompted us to consider our data from new and interesting angles, the culmination of which we feel has greatly strengthened the work. Thank you for your time.

Editor’s comments:

[Comment #1]

Limited Contribution to the Literature

Your study’s findings largely align with existing research. Please clearly articulate what new insights your study provides.

[Author action #1]

Thank you for your thoughtful feedback. We believe we have already addressed our study's novelty and unique insights. We would be grateful if you could review that part, and we’re happy to further clarify or expand on any points if needed. Also, we re-evaluated the novelty of this study, which we believe primarily consists of the following three aspects:

1.Measurement under conditions close to daily life: Unlike other studies that use prescribed footwear, our study measured participants in their habitual shoes, collecting data that more accurately represents natural, everyday walking styles. Furthermore, many previous studies have conducted measurements on treadmills, which differ significantly from actual walking conditions. By performing measurements in real-life conditions, our study is able to provide practical insights that more realistically reflect individual walking patterns.

2.High-precision data collection of daily movement using IMS: Instead of traditional motion capture systems or treadmill-based experiments, we utilized the In-Shoe Motion Sensor (IMS) system, allowing measurements in an environment closer to everyday life. This enables a realistic evaluation of the impact of wearing high heels in typical environments outside the laboratory, offering valuable feedback applicable to real-world conditions.

3.Practical application perspective: The data collection method using IMS serves as foundational data that can be leveraged to evaluate fall risks in everyday walking and provide feedback for designing comfortable and safe footwear. This data can be applied in the future to develop more refined footwear suited for everyday walking and create effective gait training programs, adding further value to this study.

The specific parts demonstrating the novelty are as follows:

(The Introduction)

Research on the effects of high-heeled shoes on gait parameters has significantly progressed. However, many evaluations have been conducted using a treadmill with participants walking at a constant speed [14]. Treadmill walking occurs under flat, steady-speed conditions, which differs from everyday walking environments [15]. For example, because a treadmill maintains a fixed speed, participants cannot walk at their natural pace, potentially suppressing their typical gait patterns [16,17]. Additionally, on a treadmill, individuals often need to adjust their posture and gait mechanics to match the treadmill’s motion in order to maintain visual and balance cues, which is known to influence gait biomechanics [15]. In certain studies, participants wore unfamiliar shoes that may not accurately reflect their usual gait patterns in daily life [11,14]. Shoes that are unfamiliar or uncomfortable for an individual often do not fit well with the person’s foot shape or gait style, potentially affecting the gait biomechanics [18]. Specifically, when the shoe’s size or shape does not align with the foot, it may cause foot pain and fatigue, while also reducing stability and increasing the risk of falls during walking [19]. Therefore, conditions different from those in daily life may alter an individual’s natural gait patterns and may not accurately reflect habitual walking [11].

As a more suitable evaluation method, Inertial Measurement Units (IMUs) can be used. IMUs can estimate three-dimensional movement from angular velocity and acceleration [20,21]. Due to their low cost, small size, and ease of use, they have been widely used to evaluate daily life walking in recent years [22]. By using the In-Shoe Motion Sensor system (IMS), which measures the acceleration and angular velocity of the feet using IMUs and calculates walking parameters, it is possible to obtain walking data comparable to daily life more accurately than conventional studies [23].

Line numbers where the changed text appears: 81-102

(The introduction for Discussion)

This study aimed to evaluate the effect of high-heeled shoes on gait parameters in healthy adult women using an IMS system. In previous studies, participants often wore unfamiliar shoes and measurements were commonly conducted on a treadmill at a constant speed, which may not accurately reflect natural gait patterns in real-life settings. In contrast, this study utilized a compact and user-friendly IMS to collect gait data while participants wore their own familiar shoes. This approach allowed for accurate collection in a manner closer to daily life conditions, providing foundational data for assessing the effects of footwear under more realistic circumstances.

Line numbers where the changed text appears: 221-227

(Conclusion)

Because this study collected gait data under conditions similar to those of daily life, it provides data suitable for practical applications and may contribute to future research focused on evaluating gait in everyday scenarios.

Line numbers where the changed text appears: 319-321

[Comment #2]

Sample Size and Power Issues

Seventeen participants are insufficient for drawing meaningful conclusions. Even though you state a power calculation, the rationale for choosing an effect size of 0.62 is unclear.

If increasing the sample size is not feasible, discuss this limitation more transparently in the manuscript.

[Author action #2]

Thank you for your valuable comments. We have revised the limitations section to clearly describe the constraints of the sample size and its impact on statistical power. Additionally, we have removed the mention of the sample size from the participants section. We have learned that if increasing the sample size is not feasible in future research, it is important to discuss this limitation more transparently, and we will keep this in mind moving forward. Thank you again for your insights. We appreciate your time in reviewing our revisions at your convenience.

The changed text is highlighted in yellow as follows:

There are limitations in this study; the participants were limited to healthy adult women from the same university, which restricts the generalizability of our findings. To apply these results to other age groups, sexes, or individuals with specific health conditions, further studies with more diverse participants are necessary. Second, the sample size in this study was limited to 17 participants, which poses constraints on statistical robustness and the power to detect effects. In particular, to adequately assess the impact of individual differences, it is essential to include a larger number of participants. Future studies should increase the sample size to enhance statistical robustness and improve the power to detect effects. Finally, we were unable to capture the movements of the hip and knee joints, making it challenging to fully assess how these joints affect gait mechanics, particularly balance and posture control. Specifically, it remains unclear to what extent high-heeled shoes stress on these joints under near-daily conditions and how this stress might increase the risk of falls or lead to long-term joint disorders. Future research should incorporate devices that measure the movements of these joints to provide a more comprehensive evaluation of the impact of different footwear types on gait and balance.

Line numbers where the changed text appears: 293-297

[Comment #3]

Methodology Needs More Justification

Shoe Selection: Your definition of high-heeled shoes is vague. Standardizing heel height and shoe type would improve reproducibility.

Data Collection: The justification for measuring only three steps at a time is weak. Why not use a continuous gait analysis? Provide references that support your approach.

[Author action #3]

Thank you for your insightful comment. The definition of high-heeled shoes used in our study is based on previous research, which we have cited in the manuscript. We would appreciate it if you could review those references for a better understanding of our approach.

Previous research;

Cronin NJ. The effects of high heeled shoes on female gait: A review. Journal of Electromyography and Kinesiology. 2014;24(2):258-263. doi: 10.1016/j.jelekin.2014.01.004

About data collection, thank you for your comment. Previous studies have shown that sufficient reliability can be obtained with measurements of three steps, and this approach is particularly effective for evaluating gait parameters. Additionally, to ensure the reliability of the gait data, we perform measurements for three steps, and if the measurement is not successful, we repeat the process of measuring three steps to accurately collect the data.

Previous research;

1. Fukushi K, Huang C, Wang Z, Kajitani H, Nihey F, Nakahara K. On-line algorithms of stride-parameter estimation for in-shoe motion-sensor system. IEEE Sensors Journal. 2022;22:9636–9648. doi: https://doi.org/10.1109/JSEN.2022.3164057

2. Huang C, Fukushi K, Wang Z, Nihey F, Kajitani H, Nakahara K. An algorithm for real time minimum toe clearance estimation from signal of in-shoe motion sensor. Annual International Conference of the IEEE Engineering in Medicine and Biology Society. 2021: 6775–6778. doi: https://doi.org/10.1109/EMBC46164.2021.9629875

3. Huang C, Fukushi K, Wang Z, Nihey F, Kajitani H, Nakahara K. Method for estimating temporal gait parameters concerning bilateral lower limbs of healthy subjects using a single in-shoe motion sensor through a gait event detection approach. Sensors (Basel). 2022;22(1):351. doi: https://doi.org/10.3390/S22010351

[Comment #4]

Weak Biomechanical Explanation

The discussion lacks depth in explaining why high heels alter gait parameters. Consider integrating research on muscle activation, joint mechanics, and kinetic changes caused by high heels.

[Author action #4]

Thank you for your valuable feedback. We have revised the following sections to incorporate a more detailed biomechanical perspective. By adding supplementary explanations to clarify key points, we have aimed to enhance the reader’s understanding.

The changed text is highlighted in yellow as follows:

Additionally, when wearing high heels, the reduced base of support during walking leads to increased muscle activity to maintain balance, which in turn tends to increase energy expenditure. To minimize this increase in energy expenditure, it has been shown that walking speed and stride length are reduced as a strategy.

Line numbers where the changed text appears: 239-242

This reduction in ankle push-off may decrease the leg swing speed, which is essential for the double-pendulum effect during walking [37]. In the double-pendulum effect, an increased hip swing speed results in the relative lifting of the lower leg, consequently increasing foot clearance. This occurs as the hip and knee joints work in coordination, with the hip’s movement influencing the lower leg’s motion. Specifically, a rapid change in the hip angle causes the lower leg to rise, allowing the foot to clear obstacles before making contact with the ground. In the heel group, however, a reduction in leg swing speed leads to insufficient knee flexion in the double-pendulum mechanism, causing the lower leg to remain relatively low and potentially resulting in reduced foot clearance.

Line numbers where the changed text appears: 261-269

[Comment #5]

Please describe the calibration protocol for this device; I assume that is completed before data collection. A couple of concerns with the use of IMUs is 1) data collection interference depending on laboratory conditions, eg, metal, etc., and 2) data drift. Were these issue that you were aware and could evaluate? Please clarify this.

[Author action #5]

Thank you for your insightful comments. As stated in previous studies, this device ensures the accuracy of data collection through real-time correction. By applying this real-time correction, continuous adjustments can be made, allowing for the elimination of experimental environmental errors as much as possible and enabling more precise data measurements. We appreciate your understanding. Additionally, since the experimental environment was designed to closely resemble daily life conditions, it was not possible to achieve complete control. However, as mentioned in prior research, the application of two algorithms—the Three-dimensional Zero-Velocity-Update (3D ZUPT) algorithm and the On-line Stride Segmentation algorithm—ensures measurement accuracy. The on-line stride segmentation algorithm enables stable foot-flat detection using the foot angle, thereby achieving accurate stride segmentation. Meanwhile, the 3D ZUPT algorithm effectively corrects drift errors, allowing for precise stride parameter calculations within the IMS device. By working in conjunction, these algorithms compensate for measurement errors caused by environmental factors and data drift, ensuring highly reliable data collection. Additionally, I have incorporated further explanations into the revised manuscript. I would appreciate it if you could review it at your convenience.

This IMS maintains measurement accuracy by applying real-time correction through two algorithms: the Three-dimensional zero-velocity-update (3D ZUPT) algorithm and the On-line stride segmentation algorithm. The on-line stride segmentation algorithm enables stable foot-flat detection using the plantar angle, ensuring precise stride segmentation. Additionally, the 3D ZUPT algorithm effectively compensates for drift errors, allowing for accurate stride parameter computation within the IMS device [25].

The changed text is highlighted in yellow as follows: 140-145

[Comment #6]

please provide a scale with a range of effect size that indicate the strength of the effect size, i.e., its clinial significance.; reference this scale

[Author action #6]

Thank you for your suggestion. We have revised the manuscript to include a scale that indicates the strength of the effect size and its clinical significance, along with the appropriate reference. Please review the updated section at your convenience.

The changed text is highlighted in yellow as follows:

The effect size was evaluated using Cohen’s d to assess the magnitude of the differences between groups, with thresholds defined as Small (0.2–0.49), Medium (0.5–0.79), and Large (≥ 0.8) [28].

Line numbers where the changed text appears: 193-195

---

## [Decision Letter · Decision Letter 2]

PONE-D-24-32099R2The effects of high-heeled shoes on gait parameters in healthy adult womenPLOS ONE

Dear Dr. Aoyama,

Thank you for submitting your manuscript to PLOS ONE. After careful consideration, we feel that it has merit but does not fully meet PLOS ONE’s publication criteria as it currently stands. Therefore, we invite you to submit a revised version of the manuscript that addresses the points raised during the review process.

We look forward to receiving your revised manuscript.

Kind regards,

Qichang Mei

Academic Editor

PLOS ONE

Journal Requirements:

Reviewers' comments:

Reviewer's Responses to Questions

**Comments to the Author**

1. If the authors have adequately addressed your comments raised in a previous round of review and you feel that this manuscript is now acceptable for publication, you may indicate that here to bypass the “Comments to the Author” section, enter your conflict of interest statement in the “Confidential to Editor” section, and submit your "Accept" recommendation.

Reviewer #1: All comments have been addressed

Reviewer #3: All comments have been addressed

2. Is the manuscript technically sound, and do the data support the conclusions?

Reviewer #1: Yes

Reviewer #3: Yes

3. Has the statistical analysis been performed appropriately and rigorously? 

Reviewer #1: Yes

Reviewer #3: Yes

4. Have the authors made all data underlying the findings in their manuscript fully available?

Reviewer #1: Yes

Reviewer #3: Yes

5. Is the manuscript presented in an intelligible fashion and written in standard English?

Reviewer #1: Yes

Reviewer #3: Yes

6. Review Comments to the Author

Reviewer #1: Thank you for addressing the previous questions/comments/concerns. This revision has improved the manuscript. However, I have noted some additional corrections in word choice and tense. See pdf for specific comments which were made in the yellow-highlighted portion of the submission.

Reviewer #3: Thank you for your thoughtful and comprehensive responses to the previous reviewer comments. It is clear that you have invested considerable effort into addressing the concerns raised, and the revised manuscript reflects significant improvement. I appreciate the clarity with which you justified the methodological decisions and enhanced the biomechanical discussion.

Below are a few final suggestions and observations that may help further strengthen your manuscript prior to publication:

Theoretical Context and Terminology:

The revised introduction and discussion more clearly articulate the novelty of the study. However, to improve cohesion, I recommend defining terms such as “habitual shoes,” “natural gait,” and “real-life conditions” early on and using them consistently throughout the manuscript.

Methodology Clarity:

While your explanation for using a three-step gait measurement approach is now supported by literature, the rationale could benefit from slightly deeper elaboration. Briefly explaining why this method was chosen over continuous gait analysis (e.g., sensor limitations, data precision, ecological validity) would enhance transparency.

Standardization of High-Heeled Shoe Definition:

Although you referenced existing literature to define high-heeled shoes, consider including a more specific numerical range (e.g., heel height in centimeters) within your methods section to improve reproducibility for future studies.

Effect Size Interpretation:

The inclusion of Cohen’s d scale is appropriate and appreciated. However, to enhance interpretability for clinical or applied readers, you might briefly comment on how the observed effect sizes relate to meaningful changes in gait mechanics (e.g., stride length, foot clearance).

Calibration Protocol and Sensor Accuracy:

Your explanation of the 3D ZUPT and stride segmentation algorithms is technically sound. One suggestion would be to clarify explicitly whether a manual calibration procedure was performed prior to data collection, as this is a common step in IMU-based research.

These are minor revisions and do not necessitate reanalysis or structural changes. I believe your manuscript is now close to being ready for publication and makes a valuable contribution to the field of gait analysis and wearable motion sensor applications.

Thank you again for your careful revisions and scholarly engagement.

7. PLOS authors have the option to publish the peer review history of their article (what does this mean? ). If published, this will include your full peer review and any attached files.

**Do you want your identity to be public for this peer review?** For information about this choice, including consent withdrawal, please see our Privacy Policy .

Reviewer #1: No

Reviewer #3: **Yes: ** MURAT ALİ ÇINAR

---

## [Author Response · Author response to Decision Letter 3]

26 May 2025

Point-by-point response

We are very grateful to the editors for providing such valuable, thoughtful, and thorough comments. The comments and questions raised have prompted us to consider our data from new and interesting angles, the culmination of which we feel has greatly strengthened the work. This manuscript has not been published or presented elsewhere and is not under consideration by any other journal. All study participants provided informed consent, and the study design was approved by the appropriate ethics review board. We have read and understood your journal’s policies and believe that neither the manuscript nor the study violates any of these policies.

Thank you for your time.

[Comment #1]

Theoretical Context and Terminology:

The revised introduction and discussion more clearly articulate the novelty of the study. However, to improve cohesion, I recommend defining terms such as “habitual shoes,” “natural gait,” and “real-life conditions” early on and using them consistently throughout the manuscript.

[Author action #1]

Thank you very much for your valuable feedback. As you rightly pointed out, clearly defining key terms at the beginning of the manuscript is essential for enhancing the clarity and consistency of the overall text. Therefore, we have added definitions for the terms “habitual shoes,” “natural gait,” and “real-life conditions” in the Introduction section and ensured that these terms are used consistently throughout the manuscript. We believe these revisions have improved the theoretical foundation and overall readability of the paper. The revised sections are as follows:

In this study, real-life conditions refer to natural walking situations that closely resemble daily life, without the use of treadmills or experimental footwear. Natural gait refers to an individual’s inherent walking pattern under conditions without external constraints or artificial instructions, and without any intentional modification of their gait.

Line numbers where the changed text appears: 184-187

Accordingly, the habitual shoes used in this study refer to the footwear that the participants considered to be the ones they wore most frequently during daily activities, such as commuting and shopping.

Line numbers where the changed text appears: 161-165

[Comment #2]

Methodology Clarity:

While your explanation for using a three-step gait measurement approach is now supported by literature, the rationale could benefit from slightly deeper elaboration. Briefly explaining why this method was chosen over continuous gait analysis (e.g., sensor limitations, data precision, ecological validity) would enhance transparency.

[Author action #2]

Thank you very much for your valuable feedback. We would like to respond to this matter from the perspective of data precision. In order to ensure the reliability of the three-step gait measurement in this study, we compared the RMSE values reported in a previous study (Matsumura et al., 2022, IEEE) using the same three-step gait approach with the intergroup differences observed in our research. As a result, the intergroup differences exceeded the RMSE in most variables, suggesting that the measurement method used in our study possesses a certain degree of reliability. We believe that this revision is the most appropriate way to demonstrate the reliability of our measurement approach among the options we considered. We hope this measurement method will be adopted by more researchers in future studies. We kindly ask for your understanding and confirmation on this point. The revised sections are as follows:

In a previous study [25], the RMSE for stride length and walking speed based on measurements taken over three steps was reported as 0.042 and 0.071, respectively. In contrast, the differences between the HG and SG in the present study were 0.1346 for stride length and 0.47 for walking speed, both of which substantially exceeded the reported RMSE values. Therefore, these differences are considered to exceed the range of measurement error and can be interpreted as practically meaningful and statistically significant group differences.

Line numbers where the changed text appears: 246-251

In a previous study [25], the RMSE for foot height was reported to be 0.017. In contrast, the difference between the HG and SG observed in this study was 0.0464, which substantially exceeds the RMSE. Therefore, the difference in foot clearance can be considered to exceed the margin of measurement error and represents a practically and statistically significant intergroup difference.

Line numbers where the changed text appears: 270-273

In a previous study [25], the RMSE for toe angle during the swing phase was reported to be 2.071. In contrast, the difference between the HG and SG observed in this study was 1.91, which is approximately the same as the RMSE. Therefore, it is possible that the observed difference in toe angle may not be entirely free from the influence of measurement error.

Line numbers where the changed text appears: 306-310

Finally, for the toe angle, the difference between groups was comparable to the RMSE, making it difficult to conclusively determine that the observed difference was statistically meaningful. Therefore, future studies should adopt more precise measurement techniques to verify the reliability of this difference.

Line numbers where the changed text appears: 324-327

[Comment #3]

Standardization of High-Heeled Shoe Definition:

Although you referenced existing literature to define high-heeled shoes, consider including a more specific numerical range (e.g., heel height in centimeters) within your methods section to improve reproducibility for future studies.

[Author action #3]

Thank you very much for your valuable feedback. In response to your comment, we have revised the Methods section to explicitly state the heel height in centimeters. In this study, we defined high-heeled shoes as those with a heel height of 3 cm or more. This definition was based on previous studies that examined gait characteristics in young adult females (e.g., Lee, 2014; Park, 2009), in which shoes with heel heights of 3 cm or greater were used. By specifying a numerical range in centimeters, we aimed to enhance the reproducibility and clarity of the operational definition of high-heeled shoes. We appreciate your thoughtful review and hope the revision addresses your concern. The changed text is highlighted in yellow as follows:

Based on previous studies that used shoes with a heel height of 3 cm or more as high-heeled footwear, the present study defined high-heeled shoes as those with a heel height of 3 cm or greater.

Line numbers where the changed text appears: 161-165

(Previous research)

Lee CR. The Effects of Lower Extremity Angle According to Heel-height Changes in Young Ladies in Their 20s during Gait. Journal of Physical Therapy Science. 2014;26(7):1055–1058. doi: 10.1589/jpts.26.1055

Park JJ. A Comparative Analysis on Changes of Foot Pressure by Shoe Heel Height during Walking. Korean Journal of Applied Biomechanics. 2009;19(4):771–778. doi: 10.5103/KJSB.2009.19.4.771

[Comment #4]

Effect Size Interpretation:

The inclusion of Cohen’s d scale is appropriate and appreciated. However, to enhance interpretability for clinical or applied readers, you might briefly comment on how the observed effect sizes relate to meaningful changes in gait mechanics (e.g., stride length, foot clearance).

[Author action #4]

Thank you for your insightful comment regarding the interpretation of effect sizes. In response, we have added a brief explanation of how the observed effect sizes relate to meaningful changes in gait mechanics, such as stride length and foot clearance. The changed text is highlighted in yellow as follows:

Regarding spatial parameters such as walking speed, stride length, and swing speed, all were significantly reduced in the high-heeled group, with large to moderate effect sizes: Cohen’s d = -1.09 for stride length and walking speed, and Cohen’s d = -0.75 for swing speed. These results strongly suggest that wearing high-heeled shoes has a substantial impact on propulsion and walking efficiency. Generally, wearing high heels narrows the base of support during walking, prompting individuals to adopt a safer and more stable gait pattern [31,32].

Line numbers where the changed text appears: 240-246

A very large effect size for foot clearance (Cohen’s d = -1.09) and a large effect size for foot height (Cohen’s d = -0.90) strongly suggest that foot clearance was more reduced in the high-heeled group.

Line numbers where the changed text appears: 267-269

This change demonstrated a moderate effect size (Cohen’s d = 0.54), suggesting that individuals wearing high heels may adjust foot positioning to enhance stability.

Line numbers where the changed text appears: 296-298

[Comment #5]

Calibration Protocol and Sensor Accuracy:

Your explanation of the 3D ZUPT and stride segmentation algorithms is technically sound. One suggestion would be to clarify explicitly whether a manual calibration procedure was performed prior to data collection, as this is a common step in IMU-based research.

[Author action #5]

Thank you for your comment regarding the calibration protocol and sensor accuracy. In this study, we did not perform manual calibration, as we relied on the ZUPT and stride segmentation algorithms for accurate measurements. We have now clarified this point in the revised manuscript. The changed text is highlighted in yellow as follows:

Due to the implementation of these algorithms, no manual calibration of the IMU was conducted in this study.

Line numbers where the changed text appears: 141-142

---

## [Editor Report · Decision Letter 3]

The effects of high-heeled shoes on gait parameters in healthy adult women

PONE-D-24-32099R3

Dear Dr. Aoyama,

We’re pleased to inform you that your manuscript has been judged scientifically suitable for publication and will be formally accepted for publication once it meets all outstanding technical requirements.

Kind regards,

Qichang Mei

Academic Editor

PLOS ONE
---

## [Editor Report · Acceptance letter]

PONE-D-24-32099R3

PLOS ONE

Dear Dr. Aoyama,

I'm pleased to inform you that your manuscript has been deemed suitable for publication in PLOS ONE. Congratulations! Your manuscript is now being handed over to our production team.

Kind regards,

on behalf of

Professor Qichang Mei

Academic Editor

PLOS ONE